# Formation Patterns of Mediterranean High-Mountain Water-Bodies in Sierra-Nevada, SE Spain

Jose Luis Diaz-Hernandez [1,*] and Antonio Jose Herrera-Martinez [2]

1    IFAPA Camino de Purchil, Área de Recursos Naturales, Consejería de Agricultura, Pesca y Desarrollo Rural, Junta de Andalucía, Apartado 2027, 18080 Granada, Spain
2    ENSN, Área de Uso Público, Consejería de Medio Ambiente y Ordenación del Territorio, Junta de Andalucía, Pinos Genil, 18191 Granada, Spain; antonioj.herrera@juntadeandalucia.es
*    Correspondence: jldiaz.terra@gmail.com

**Abstract:** At present, there is a lack of detailed understanding on how the factors converging on water variables from mountain areas modify the quantity and quality of their watercourses, which are features determining these areas' hydrological contribution to downstream regions. In order to remedy this situation to some extent, we studied the water-bodies of the western sector of the Sierra Nevada massif (Spain). Since thaw is a necessary but not sufficient contributor to the formation of these fragile water-bodies, we carried out field visits to identify their number, size and spatial distribution as well as their different modelling processes. The best-defined water-bodies were the result of glacial processes, such as overdeepening and moraine dams. These water-bodies are the highest in the massif (2918 m mean altitude), the largest and the deepest, making up 72% of the total. Another group is formed by hillside instability phenomena, which are very dynamic and are related to a variety of processes. The resulting water-bodies are irregular and located at lower altitudes (2842 m mean altitude), representing 25% of the total. The third group is the smallest (3%), with one subgroup formed by anthropic causes and another formed from unknown origin. It has recently been found that the Mediterranean and Atlantic watersheds of this massif are somewhat paradoxical in behaviour, since, despite its higher xericity, the Mediterranean watershed generally has higher water contents than the Atlantic. The overall cause of these discrepancies between watersheds is not connected to their formation processes. However, we found that the classification of water volumes by the manners of formation of their water-bodies is not coherent with the associated green fringes because of the anomalous behaviour of the water-bodies formed by moraine dams. This discrepancy is largely due to the passive role of the water retained in this type of water-body as it depends on the characteristics of its hollows. The water-bodies of Sierra Nevada close to the peak line (2918 m mean altitude) are therefore highly dependent on the glacial processes that created the hollows in which they are located. Slope instability created water-bodies mainly located at lower altitudes (2842 m mean altitude), representing tectonic weak zones or accumulation of debris, which are influenced by intense slope dynamics. These water-bodies are therefore more fragile, and their existence is probably more short-lived than that of bodies created under glacial conditions.

**Keywords:** hillside instability; landslides; mountain streams; overdeepening; water-bodies of Mediterranean mountains

## 1. Introduction

Ephemeral waters, characterised by periodic flows in space and time, represent over half the total discharge of the worldwide river network [1], with an increase forecast due to climatic changes and greater demand for water resources [2,3]. Increasing research on intermittent river ecology has documented the importance of the meteorological, geological and land-cover components of these ecosystems on the structure of ecological communities [4–10], but mechanisms controlling flow permanence remain poorly understood. We must also consider that water bodies in mountain areas are the prime exporters of

solutes to the surroundings after winter snowpack depletion and the summer dry season in the Mediterranean area (Figures 1 and 2). As yet, little attention has been paid to the circumstances coming together in this process [11–15], but this analysis requires a definition of the present state of the environment, which in itself is a question with significant gaps in our understanding.

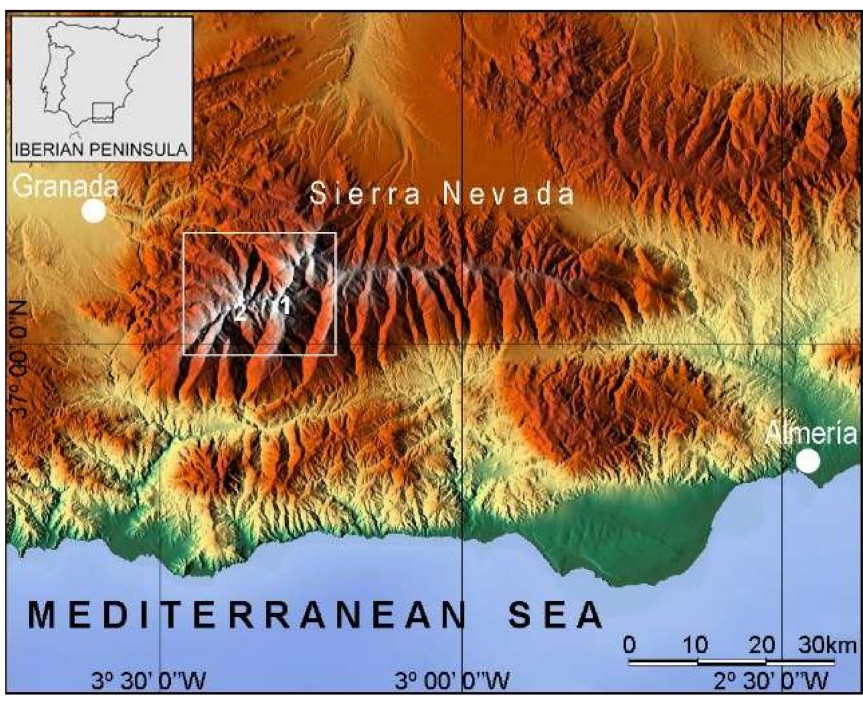

**Figure 1.** Geographic situation of the Sierra Nevada Massif: the white box shows the area where the studied water-bodies are located. Numbers 1 and 2 indicate the Mulhacén and Veleta peaks.

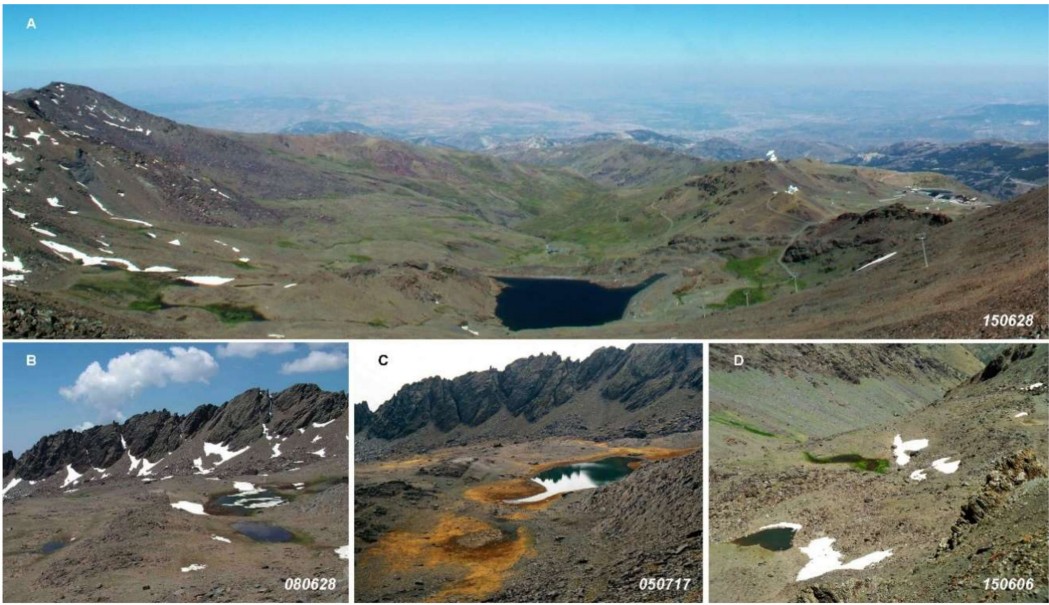

**Figure 2.** Some observable relations between water-bodies and adjoining watercourses: (**A**) headwaters of the Dílar stream, where Las Yeguas reservoir (centre of image) is supplied by small natural water bodies (Lagunas de la Virgen, left); (**B**) headwaters of the Rioseco stream, where the Rioseco water-body is located, towards the end of thaw and (**C**) in summer of a very dry year; and (**D**) two waterbodies at the headwaters of Valdeinfierno towards the end of thaw, in which the lower is permanent, fed by groundwater from the upper, which is temporary. The numbers in lower right corners indicate the date of each photograph (year, month, day).

Sierra Nevada contains numerous small, shallow, natural water-bodies scattered throughout the highest parts of this massif [16]. Located in the SE of the Iberian Peninsula (Figure 1), these biotopes basically exist because of a supply of persistent snow lying for over seven months per year on the high peaks of the western sector of the massif. However, this is not a sufficient condition to explain the origin of the hollows containing them. The literature has many examples illustrating the different causes of the presence of water-bodies, frequently observed from a risk-management viewpoint [17–20], where the risks could compromise the biodiversity linked to these water-bodies [21]. The suggestion is that these morphologies could have causes other than glacier dynamics [22], such as tectonic activity, in particular when they are located at more southerly latitudes.

Indeed, the features of quaternary glacial activity are well known in this context [23,24], although there are none in the present interglacial phase [25,26]. Furthermore, the steep inclines of the slopes of Sierra Nevada facilitate abundant cases of landslide, which frequently affect infrastructures and properties located in lower areas to varying degrees of magnitude and range. Together with the aforementioned action of glaciers, these phenomena complement and define the variety of causes of the different water-bodies in this massif. The geological, geographical and climatic contexts of Sierra Nevada notably modify these processes and can make them particularly sensitive to sharp climatic alterations. Based on realistic warming scenarios, it has been established that 75% of the glaciated area in the Alps at the end of the 20th century could disappear by the middle of the present century [27,28]. Consequently, some models based on GIS suggest that glacier retreat could uncover overdeepening areas at certain points that would be potential locations for water-body formation [29]. At such southern latitudes as Sierra Nevada, this circumstance was reached some time ago.

Moreover, landslides are gravitational collapses, often controlled by high rates of river incision forcing the hillsides to maintain the landslide threshold angle [30]. Some authors [31] considered that landslides are the main cause of denudation in tectonically active mountain ranges, of which western Sierra Nevada offers good examples according to its high uplift rate [32,33]. Additionally, in mountain belts intersecting the snow line, glacial and periglacial processes place an upper limit on altitude, relief and the development of topography regardless of the rate of tectonic processes operated [34].

Lake hollows can also form via a dam effect when a landslide blocks a river and the stream forms a water-body at the valley bottom. These are usually short-lived because of the erosive action of water on the damming material. Costa and Schuster [35] classified the most common types of natural dams in high-mountain regions as ice dams, moraine dams, landslide dams and bedrock dams. However, small water-bodies can also form in the displaced material merely by the creation of small endorheic areas due to slope accommodation.

Therefore, the main aim of this study is to classify and locate the water-bodies of Sierra Nevada according to patterns of formation including phenomena other than those due to glacial origin. This would help to provide a more complete view of the evolution of these types of Mediterranean massifs and their singularities. These objectives were obtained from geomorphological evidence. Reconnaissance in the field was carried out with the assistance of Google Earth as the most adequate remote sensing method for locating and monitoring temporary ponds and, in these circumstances, provided similar usefulness as that indicated in other cases [36]. Whether these high-mountain water-bodies have an expanding or reducing tendency, such water-bodies would pose risks and opportunities that should be considered among the priorities of the Mountain Agenda [37] and subsequent policy actions such as those adopted as Sustainable Development Goals (SDGs), built up over decades by work in all United Nations member states [38]. Consequently, this study will help to understand the role of each formation process in the creation of aquatic spaces of ecological importance in this context, their relations with their streams feeding them or fed by them, and the possible response of these spaces to changes.

## 2. Materials and Methods

### 2.1. Study Area

The study area is located in the central-western part of the Sierra Nevada massif (SE of Iberian Peninsula, Figure 1), which extends to approximately 30 km maximum length between the geographical coordinates 3°32′–2°50′ W–36°55′–37°10′ N, at altitudes between 2480 m asl (above sea level) and the Mulhacén peak (3479 m asl), which is the maximum altitude of the Iberian Peninsula and the third highest massif in Europe. The central part of this massif was declared a national park in 1999 due to its significant, high Mediterranean mountain plant endemism and constitutes one of the most important world reserves of plant diversity. This study did not endanger protected species.

There are few weather stations, but the mean annual precipitation recorded varied from 710 mm at the University Hostel station (2507 m asl, Granada University [39]) to 507 mm at the Capileira station (1588 m asl; Automatic Hydrological Information System of the Ministry for the Ecological Transition and the Demographic Challenge) [40]. At present, snow cover is only seasonal, between November and June, and the thaw process varies in duration, sometimes including the month of August [16]. The southern position of the massif and the Mediterranean influence cause its relative dryness, with minimal precipitation in the summer. Because of these features, the ecosystems of this massif are good examples for study from an insufficiently used viewpoint in high Mediterranean mountains.

### 2.2. Geology

The Nevado–Filábride Complex [41–43] forms the geological nucleus of Sierra Nevada. Briefly, this complex has two lithological units: the main, lower unit is the Veleta Unit, which constitutes over 90% of the outcrops in western Sierra Nevada. It consists mainly of a thick (>2 km) monotonous sequence of metamorphic rocks, essentially formed by graphite-bearing metapelite with intercalated quartzite and rare marble. This unit is presumed to be Palaeozoic (or older) in age [44]. It is overlain by the Mulhacén Unit, formed by light-coloured metasediments reaching several hundred metres thick with mica schist and quartzite in the base, followed by a sequence of mixed lithology, such as marble, calcschist, and metamorphized mafic and acid igneous rocks [45]. This complex is folded in an anticline, outcropping in the nucleus of a tectonic window surrounded by carbonated materials of the Alpujárride Complex on the national park boundary. Neo-tectonic movements are intensive on the margins of western Sierra Nevada, where uneven uplift has been caused since the early Pliocene and where the peaks are higher than 3000 m [32,33,46].

There are also more recent rocks, such as moraines, glaciofluvial deposits, rock glaciers, solifluction material, debris cones, and blockfields. They are all detritic in nature and have been accumulated as irregular deposits [25]. We also found an alteration phenomena of rocky substrates along a variable strip of peaks that accumulate the effects of yearly freeze–thaw cycles. Rock erosion is therefore high, and masses of unstable loose material are produced.

### 2.3. Geomorphology

The study area of Sierra Nevada contains a great variety of well-known glacial morphologies [23,24,47,48]: cirque glaciers, hanging glacier valleys, U-shaped valleys, moraines, striations, polished substratum, erratic blocks and sheepback rocks, some of them indicated in Figure 3. These forms confirm the presence of glaciers in Sierra Nevada during the last cold stages of the Pleistocene [49–51], although likely glaciations in Sierra Nevada were of low intensity and the cirques grow very close to the ridge line [52]. At present there are no glaciers or permanent ice masses [26] as there has been a general retreat of glaciers since 15–14 ka, followed by the formation of extensive rock glaciers [53]. These forms have gradually vanished due to two main causes: (i) an interaction between successive cold stages causes overdeepening and encroachment on the foregoing forms, and (ii) the

interglacial stages, especially severe in these latitudes, also blur these diagnostic morphologies because of increased vegetation cover and soil development facilitated by gelifraction and paedogenic clays [54,55]. These factors affect the recognition of hypothetical ancient glaciations.

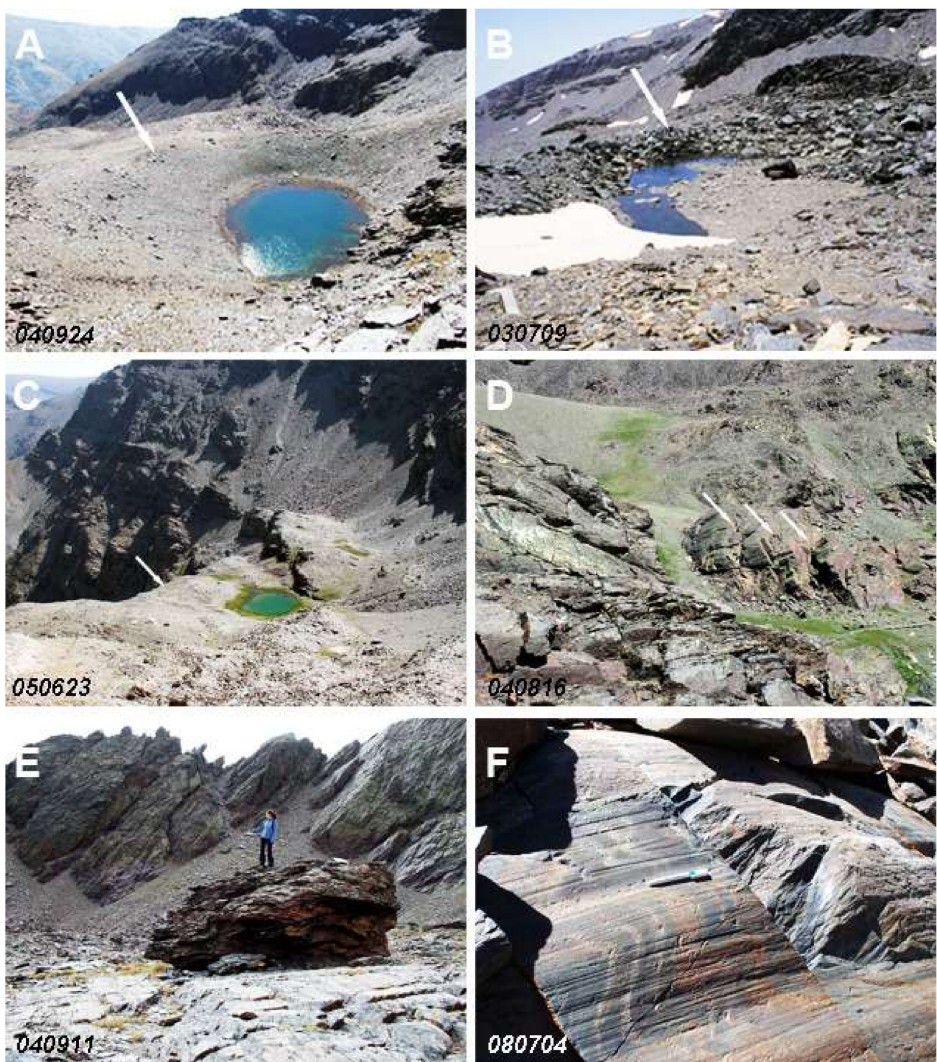

**Figure 3.** Some glacier morphologies of the area: (**A**) heterometric moraine of Laguna de Vacares, (**B**) rocky moraine and subsequent water-body near Laguna Altera, (**C**) rocky threshold of Laguna de la Mosca, (**D**) fractured and displaced sheepback rock in the Lanjarón River valley, (**E**) erratic block near Laguna de Rioseco and (**F**) rock showing glacial striations on polished bedrock in the Valdeinfierno cirque. The numbers in the lower left corner indicate the date of each photograph (year, month, day).

The cryo-Mediterranean belt (altitude range between 2480 and 3200 m asl, which also includes the snow level) shows a range of shallow water-bodies typical of this massif. Almost all remain ice-covered for long periods every year. Soils in these headwaters are in general poorly developed, and vegetation is sparse, except for the surroundings of some water-bodies, where there are "green fringes" of varied extension, locally known as "borreguiles" (Figure 4A, see below). Recent studies determined that some of the sediments filling these water-bodies are Holocene, although there is a time gradation between them [56,57].

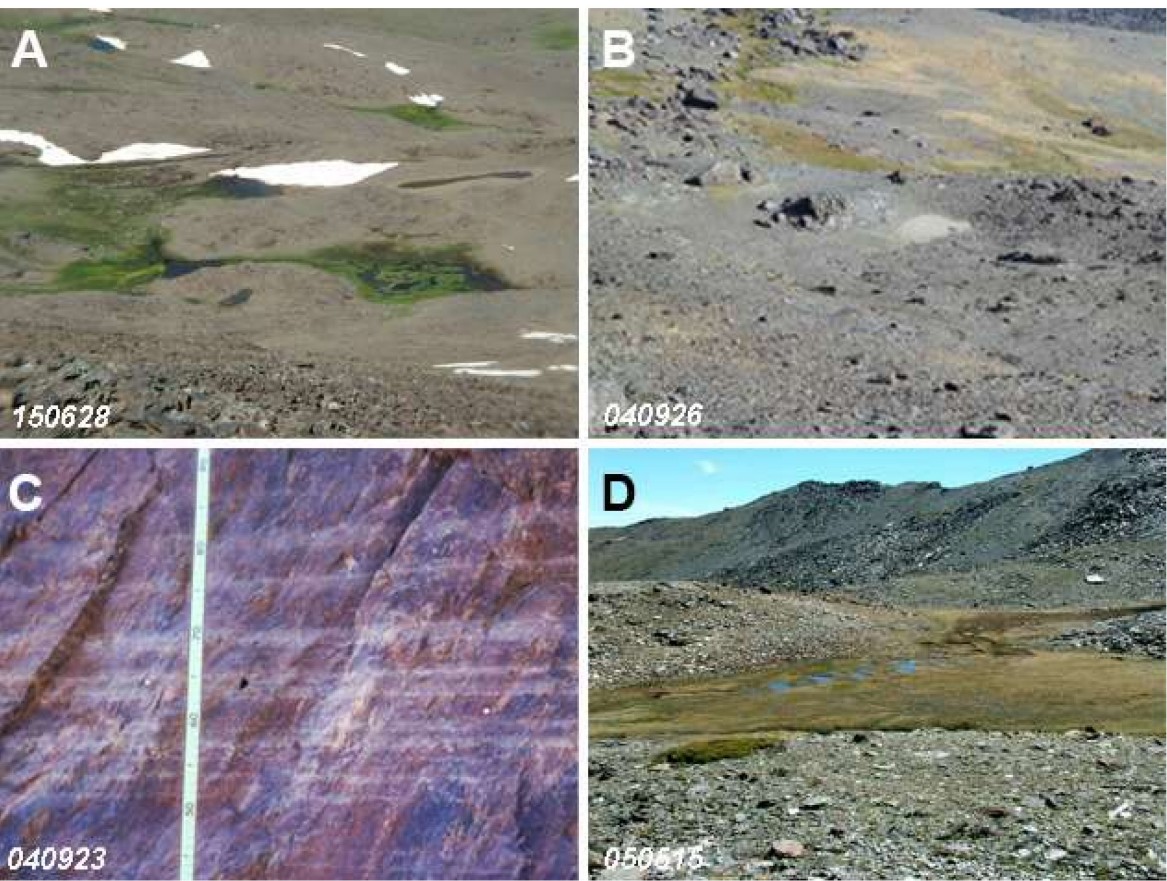

**Figure 4.** Some specific cases of (**A**) green fringes around Lagunas de la Virgen water-bodies, (**B**) sediments (whitish colour) deposited at the bottom of a small depression in the Guarnón River valley sporadically occupied by a water-body, (**C**) horizontal white bands of the maximum water level marked on a block on the edge of Laguna de Vacares and (**D**) "Stepped ponds" located in the Hoyo del Moro. The numbers in the lower left corner indicate the date of each photograph (year, month, day).

### 2.4. Methods

Most of the water-bodies are short-lived after the thaw. Aerial images of the zone therefore only record the largest water-bodies, while the smaller ones are reduced to remnants that can be distinguished by detailed field work. These field studies were carried out between the years 2000 and 2018, approximately between 1 June and 30 November without interruption, invariably after thaw and including wet and dry years. We detected and measured some temporary water-bodies thanks to the presence of sediments or thin coatings on rocks, formed in partially flooded hollows (Figure 4B) and often because of the associated green fringes (Figure 4A), visiting both cases in wetter years for confirmation. We often detected horizontal whitish lines on the rocks at the edges (Figure 4C) as indications of the maximum level of the water-surface (briefly "maximum level lines") and which we have used as indicators of "historic water-levels". Other data on the water-bodies concerned the presence or absence of glacial action, morphology of the hollows, presence or absence of green fringes, types of floors (sandy, loamy and rocky), presence of springs, and inflows or outflows. We estimated water volume by using the surface area of the water surfaces at their maximum, as mapped from highly magnified images obtained from Google Earth. This maximum was accurately assessed in the water-bodies with green fringes by markings usually found around the edge. On our field visits, we enclosed each water-body in a rectangle for which the surface was compared with those obtained by aerial photography, obtaining a good correlation (Figure 5A). In these cases, the plane of the water body was practically horizontal and distortion was minimal. Once the area was

obtained, the volume of each water-body was estimated applying the cylinder, $\frac{1}{2}$ ellipsoid, or cone formulas, depending on whether they were respectively <0.5 m, 0.5–2 m, or >2 m deep. These volumes were also compared (Figure 5B) with internal reports (measurements of around ten of water-bodies, 2009) obtained by bathymetry based on GPS stations: these methods cannot be systematically applied to all inventoried water-bodies, at least today, for logistical reasons. These comparisons allowed us to ensure a good assessment of the water contained in these water-bodies. The maximum water level used in this assessment leads us to consider that these volumes are the maximum water retained in these systems.

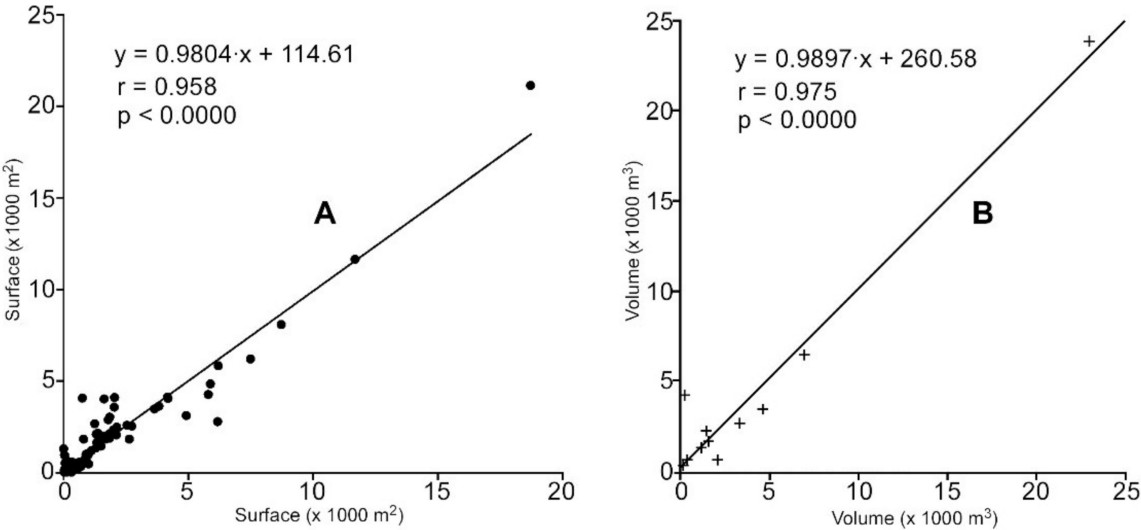

**Figure 5.** Agreement between two methods used defining (**A**) waterlogged surfaces and (**B**) volumes of water: for more detail, see the text. A statistically significant relationship between the two groups of variables analysed exists both in A and B for a confidence interval of 99%.

We represented the water-bodies on a 1:25.000 topographical basis [58], on the geomorphological map of Sierra Nevada [25], and on black and white digital orthophotos (scale 1:20.000) by the Andalusia Government (Junta de Andalucia). We also examined imagery using Google Earth Digital Globe, mainly the one recorded in summer 2015, which we compared with those of the Iberpix visualizer [59]. We likewise checked, collected and registered data such as geographical information (altitude and UTM coordinates using handheld GPS) and quantitative data (length, width and maximum water depth, using a tape-measure and a single-seat rubber boat). All this information was stored in a database, and general or detailed cartography was performed. Comparisons between some features of water-bodies and their respective watersheds were statistically analysed by the t-Student test when required, and differences were considered significant when $p < 0.05$.

At the field stage, we also photographed each water-body and recognized the associated glacier forms (see Section 2.3 and Figure 3). We considered these morphologies as indicators of the formation of hollows where these water-bodies settled. Where this is not the case, we considered a water-body not to be associated with glacial excavation. When so, we considered features associated with landslides (mainly deep slopes, head cliffs/scarps, planar surfaces, debris flows and frontal (toe) slumps), although the information on landslides derived from the literature in this area was scarce and their identification can sometimes be affected by a degree of subjectivity. We therefore classified these water-bodies of Sierra Nevada in the following recognized formation categories:

I.    Glacial (Figure 6A): (a) overdeepening and (b) moraine dams
II.   Landslides: (a) debris flow (Figure 6B) and (b) rockfall
III.  A few other cases could be anthropogenic or have an undetermined origin

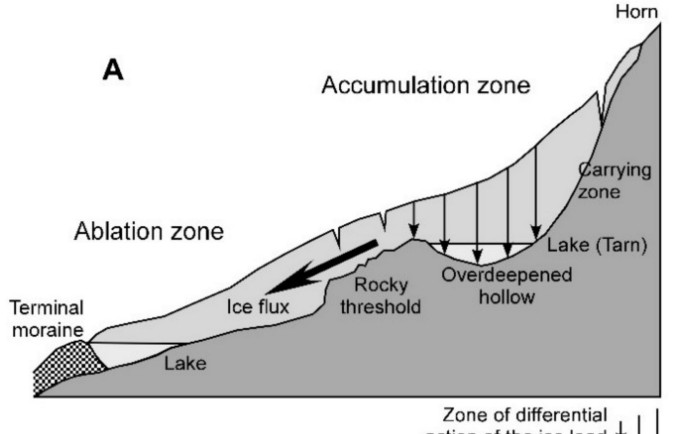 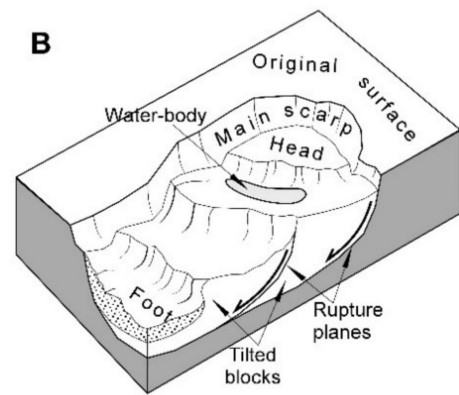

**Figure 6.** Conceptual sketch of the formation models by which the water-bodies of the Sierra Nevada can be categorized. (**A**) Alpine glacier processes, such as overdeepening and moraine dams. The lakes form when ice melts. The carrying zone is where the bedrock is removed by the action of ice and gravity and is considered the most important glacial erosive process ([60] and the references therein). (**B**) Landslides are represented by the most general case and shows some associated morphologies.

The different shapes and sizes of landslides are well known [61–64]. We therefore chose in this case to focus only on landslides (or processes caused by slope instability) which, in detail, can be related to the water volumes retained. We thus classified these processes as "debris flow" and "rock-fall"—two clearly differentiated processes that are useful in classifying the shapes studied. There are also other factors to be taken into account in landslide processes, such as size, age, and different interconnections between them, with the oldest being the most likely to cause overlapping that finally determines their dissipation. We comment below on the characteristics of some cases in order to give an overview of the main types and their similarities and differences.

Finally, those areas containing multiple small ponds, generally consisting of crescent-shaped "stepped ponds" (Figure 4D), were not included in this study because their small individual dimensions only affect upper soil horizons and, in general, constitute a special case of hillside instability.

## 3. Results

A total of 123 water-bodies were catalogued [16], mainly distributed over the western sector of Sierra Nevada between 2480 and 3200 m asl (Figure 7). The only other water body recorded in the rest of the massif is Laguna Seca, which is out of this range and therefore of secondary interest for this study. As the line of peaks lies in a clear E–W direction, we can subdivide the distribution of water-bodies by the two main watersheds of the massif because of their differing xericity—the Atlantic watershed, facing N-NW, and the Mediterranean, facing South. Water from these two watersheds was collected respectively by the Genil and Guadalfeo rivers, which are regulated by important hydraulic constructions such as the Canales (Genil River) and Rules (Guadalfeo River) reservoirs.

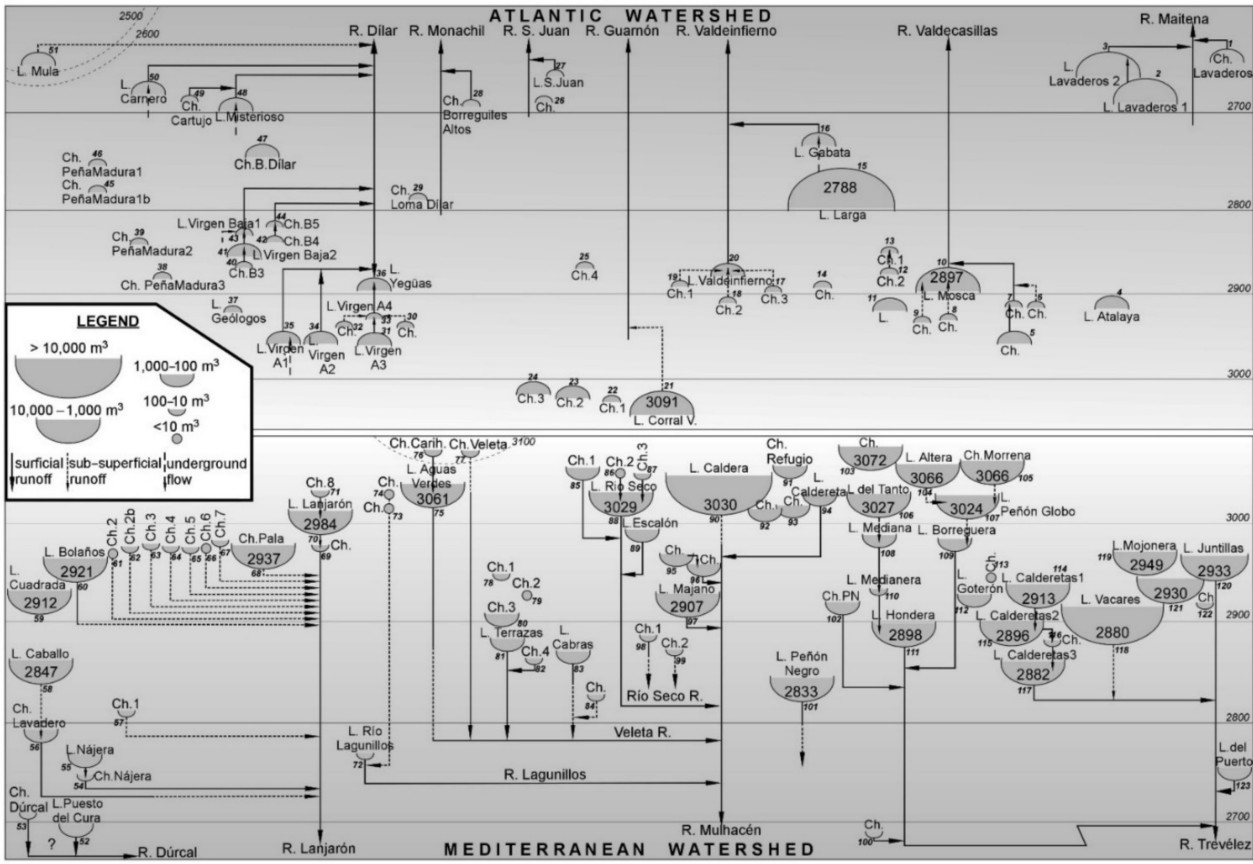

**Figure 7.** Positional relationship between the water-bodies of Sierra Nevada, ordered by watersheds and headwaters: the streams draining them as surficial/sub—surficial runoff are indicated. The larger numbers included in the larger bodies are their heights (m asl). Potential underground flows in the water-bodies (springs) and infiltrations occurring in the outflows are also shown. The water-body of Laguna Seca is not included because it is the lowest water-body (<2400 m asl). The small numbers near each water-body indicate the order.

Figure 8 shows several aerial views of some typical water-bodies. These patterns correspond to categories I (Figure 8A,B) and II (Figure 8C,D) defined in the previous section.

### 3.1. Formation Patterns and Spatial Distribution of Water-Bodies

Glacial processes (I) are mainly responsible for the formation of water-bodies in Sierra Nevada (Table 1, part A), representing 72% of the total, whereas landslides (II) caused 25%. Overdeepening phenomena are most common in the former (I), while the latter (II) are predominantly caused by debris flow. Those included under the heading "Others" (III) are anecdotal.

We found that 59% of the Sierra Nevada water-bodies lie on the Mediterranean watershed, where 45% are of glacial origin, whereas on the Atlantic watershed, 27% originated in this manner (Table 1, part A). Moraine-dammed water-bodies are scarce (5%) on both watersheds, while modelling by overdeepening is more common (40% and 22%, respectively). Slope instability (II) caused similar percentages of water-bodies on both watersheds (13% and 12%), with the most significant caused by debris flow (12% and 10%).

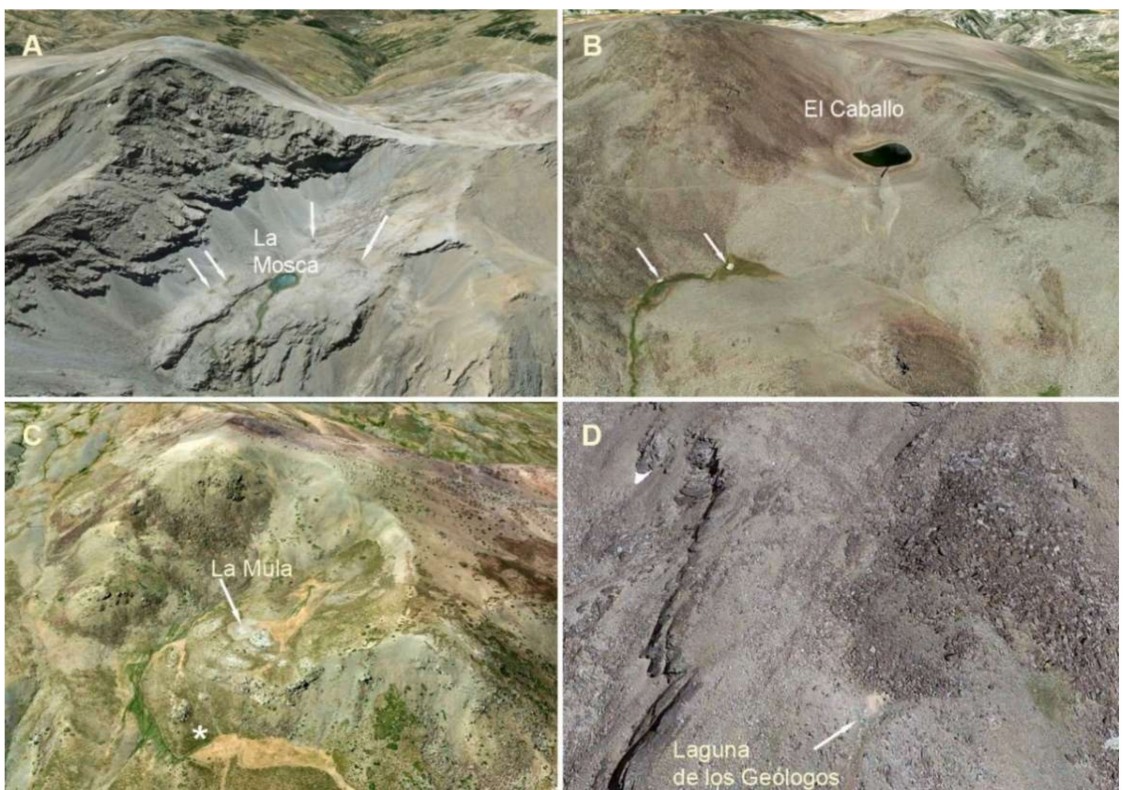

**Figure 8.** Specific cases of the different formation models according to the sketches of Figure 6: these oblique aerial views were obtained from Google Earth. The arrows point to small water-bodies. (**A**) Laguna de la Mosca (tarn) at the foot of the Mulhacén peak. (**B**) El Caballo moraine-dammed lake. (**C**) La Mula water-body caused by a landslide. The asterisk shows an overlapping landslide front over a meadow (yellowish colour). Observe the deep weathering of the substratum. (**D**) Laguna de los Geólogos showing the typical whitish sediments of the bottom of a water-body. This water-body is located at the end of two heterometric masses of rocks: the closer one, light grey in colour, contains abundant fine material, while the upper mass (dark colour) is mainly formed by rocky blocks.

**Table 1.** Main characteristics related to the different categories of water-bodies in Sierra Nevada.

| Formation Patterns | (A) Water-Bodies Number (%) | | | (B) Stored Water Volume (m³) | | | (C) Green Fringes Surface (m²) | | | (D) Mean Water Depth (m) | | |
|---|---|---|---|---|---|---|---|---|---|---|---|---|
| | Watershed | | Total Massif | Watershed | | Total Massif | Watershed | | Total Massif | Watershed | | Tot Mas |
| | Med. | Atl. | | Med. | Atl. | | Med. | Atl. | | M. | At. | |
| I Glacial | 44.7 | 26.8 | **71.5** | 125,960 | 67,923 | **193,883** | 10,1072 | 35,429 | **136,501** | | | |
| (a) Overdeep. | 39.8 | 22.0 | 61.8 | 41,643 | 65,215 | 107,858 | 100,194 | 35,429 | 135,623 | 0,9 | 0,9 | 0,9 |
| (b) Mor.-da. | 4.9 | 4.9 | 9.8 | 83,317 | 2708 | 86,025 | 878 | 0 | 878 | 5,3 | 0,9 | 3,1 |
| II Lands | 13.0 | 12.2 | **25.2** | 14,312 | 5938 | **20,250** | 47,717 | 1713 | **49,430** | | | |
| (a) Debris-fl. | 12.2 | 10.1 | 22.8 | 14,025 | 5679 | 19,704 | 47,626 | 1713 | 49,339 | 1,1 | 0,5 | 0,8 |
| (b) Rock-fall | 0.8 | 1.6 | 2.4 | 287 | 259 | 546 | 90 | 0 | 90 | 1,4 | 1,0 | 1,1 |
| III Other | 0.8 | 2.4 | **3.3** | 69 | 860 | **929** | 0 | **540** | 540 | 0,8 | 1,2 | 1,1 |
| TOTAL | **58.5** | **41.5** | **100.0** | 140,341 | 74,721 | **215,062** | 148,788 | 37,683 | 186,471 | | | |
| | | | | % = 65 | % = 35 | | % = 80 | % = 20 | | | | |

### 3.2. Formation Patterns and Stored Water Volumes

Regarding water storage, we can clearly see (Table 1, part B) the importance of the water-bodies of glacial origin (I), which store 194,000 m³ (90% of the total) as compared to the 20,000 m³ (9%) in the water-bodies formed by landslides (II), with <1000 m³ stored in

those classified as "Other" (III). The water-bodies formed by overdeepening (Ia) contain 50% of the water stored in water-bodies throughout the massif, i.e., 10% more than the water stored by moraine-dammed water-bodies. The total water stored in water-bodies formed by rock-falls is insignificant (546 m$^3$).

All of the water stored in the water-bodies on the Mediterranean watershed amounts to 140,000 m$^3$; in other words, this watershed contains 65% of the water stored in the massif and 1.88 times the amount contained in the water-bodies of the Atlantic watershed. This percentage corresponds approximately to the water stored in the water-bodies formed by glacial action (I). However, the water stored by overdeepening (Ia) on the Atlantic watershed is 1.53 times higher than that stored on the Mediterranean watershed, and the water stored in moraine-dammed water-bodies (Ib) on the Mediterranean watershed is 31 times more than the water stored on the Atlantic watershed. The amounts of water stored by landslides (II) are more evenly distributed, although the volumes on the Mediterranean watershed are invariably greater than those on the Atlantic watershed, while the volumes stored in rock-fall water-bodies (IIb) are smaller and are similar on both watersheds. We can also observe considerable contrast between the small volumes stored in others (III) on the two watersheds.

### 3.3. Formation Patterns and Associated Green Fringe Surfaces

Table 1, part C shows that the overall surface area of green fringes is higher around the water-bodies formed by glacial action (I: 136,500 m$^2$ or 73% of the total) than around those caused by slope instability (II: 50,000 m$^2$). Additionally, the Mediterranean green fringes total approximately 150,000 m$^2$, which is 4 times more extensive than those of the Atlantic watershed. The Mediterranean has larger areas of green fringes than the Atlantic watershed around both type I and type II water-bodies. The largest green fringes areas are associated with overdeepening and secondarily with debris flow water-bodies. The green fringe areas found in type III water-bodies on both watersheds are considered insignificant. We point out that all the mountain streams studied have small dimension-associated linear green fringes, which diminish as the summer drought progresses. These green fringes were not inventoried in the present study.

### 3.4. Water Depth and Water-Body Formation Types

Stored water is deeper in the water-bodies of glacial origin (Table 1, part D). The highest average depth (5.3 m) of all the water-bodies examined was found in the moraine-dammed bodies on the Mediterranean watershed. The mean depth in overdeepening water-bodies was similar on both watersheds (0.9 m) although significantly less than in moraine-dammed bodies. The Atlantic watershed has a similar mean depth in both types of water-body.

Water depths in landslide-formed bodies are more homogeneous but invariably deeper on the Mediterranean watershed and in rock-fall bodies. The shallowest water was found in Atlantic bodies caused by debris flow. The shallowest water-bodies were those of type III, although this group is rather insignificant in number.

## 4. Discussion

The data in Table 1 give an overview of the geomorphological processes involved in the formation of these water-bodies. An earlier study [16] showed the hydrological relevance of the Mediterranean watershed over the Atlantic. It described how the greater xericity of the Mediterranean watershed is in paradoxical contrast to the larger volume of water contained in its water-bodies, which in any case, overcomes the effects of low latitude in a Mediterranean context. We can now add that possible human activity is not involved, as shown by category III (other processes). If the effects of this category played a significant role in the formation of the water-bodies, they should be taken into consideration, but they are in fact merely irrelevant. This does not exclude the possibility of anthropic modifications (mainly construction of roads, small reservoirs and ditches) over

the last 250 years [65] or even in previous periods as the region is rich in history. Category III also suggests that the development of green fringes in the context of this study seems to be a relatively slow process, mainly associated to the water-bodies of glacial origin. The water-bodies linked to slope instability phenomena would therefore have formed in intermediate stages.

Moreover, the processes studied here, most specifically those caused by landslides, can have not only negative or devastating connotations but also significant implications for the conservation and formation of the cultural and natural heritage.

### 4.1. Water-Bodies of Glacial Origin

Throughout the massif, this group of water-bodies (Table 1, part A) is representative not only because of its number (72%) but also by being the closest to the peak line (2918 m mean height). This mean altitude has significant differences (Figure 9) to that of water-bodies formed by landslides ($t = 3.0916$, $p = 0.002$, confidence level 95%). The glacial origin of the water-bodies we have studied consists in the overdeepening and moraine dams that created the hollows where they lie. Other processes related to glacial erosion and rates are complex challenges requiring further research as they operate in mostly subglacial environments [22,66,67].

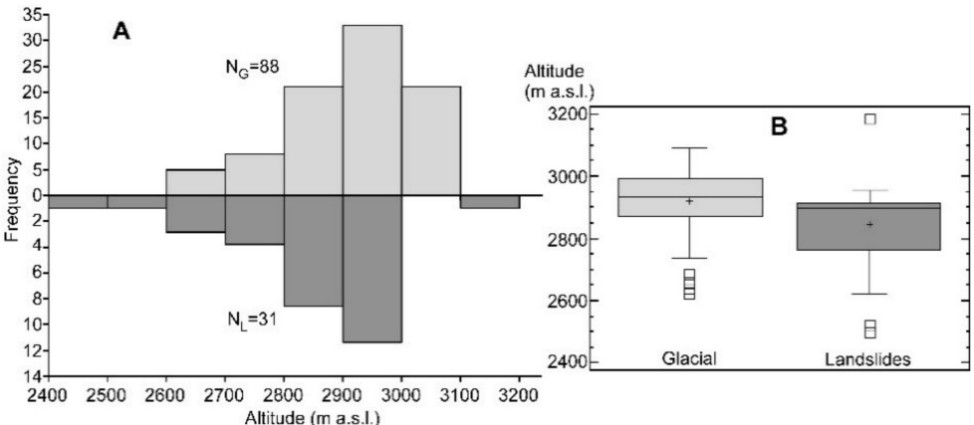

**Figure 9.** Statistical distribution of water-bodies formed by glacial and landslide processes in Sierra Nevada: (**A**) histogram showing the distribution of both formation models by altitude and (**B**) box-and-whisker plot showing differences of each formation model. N = number of elements.

Unlike other European Alpine massifs, Sierra Nevada has moraine-dammed water-bodies as high as or higher than those found in hollows only caused by overdeepening in old valleys or glacier hollows (Figure 10). This is clearly defined by both individual cases (practically all the moraine-dammed water-bodies) and by the average altitudes (2991 m vs. 2907 m asl). The moraine-dammed bodies, located at the highest parts of the massif, were formed at times of glacial minima, when glacial activity was confined to the cirques, whereas overdeepening only occurred at times of maximum glacial development.

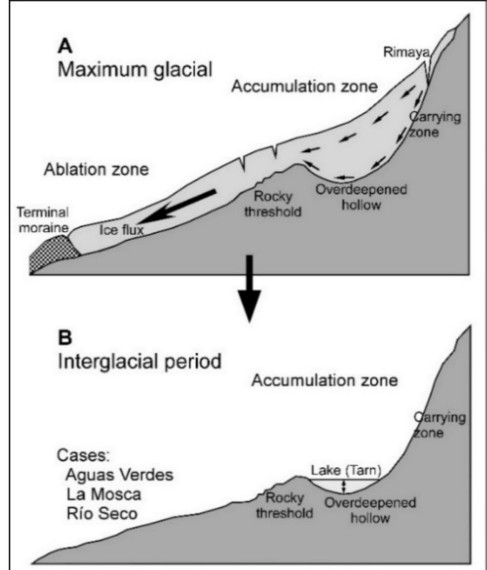
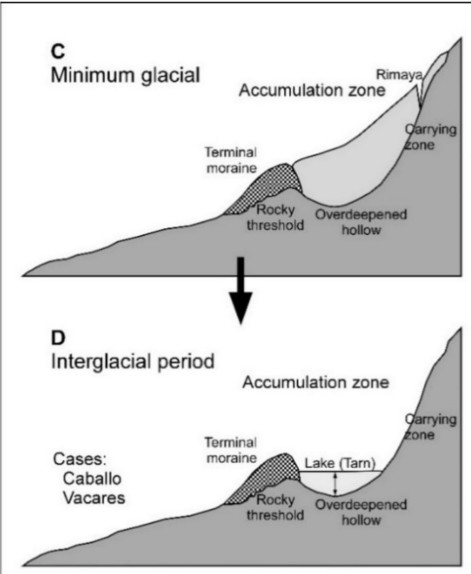

**Figure 10.** Diagrams of glaciers associated with water-bodies in Sierra Nevada: (**A**,**B**) evolution of a valley glacier from maximum development to present state and (**C**,**D**) evolution of a cirque glacier developed during an intermediate cold period until the present state.

We speculate that overdeepening basically supplies the terminal moraine with fine matter produced by abrasion of the bedrock. However, we can observe that the most developed moraines in Sierra Nevada contain rocks varying greatly in size, with a high content (46 ± 2.5%, mean value ± SE) of coarse rocks (>2 mm) as determined in a cross section from the Laguna del Caballo moraine. This coarse material would have been carried from the surrounding reliefs and would obviously be more "static" under glacial meltwater flow, tending to remain where deposited, unlike fine material resulting from abrasion.

From a morphological viewpoint, the water-bodies associated with moraines are deeper (Table 1, part D). Therefore, just 10 of these water-bodies hold a similar volume of water to that in the 62 bodies caused by overdeepening (Table 1, part B). This can be explained basically because the moraine bars are superposed on the thresholds of overdeepening hollows, thus increasing their initial volume (Figure 10D).

We should point out that the volume of water retained in moraine-dammed water-bodies has little effect on the development of the associated green fringes (Table 1, part C). In this sense, the overdeepening water-bodies on the Mediterranean watershed are much more efficient than those on the Atlantic watershed, as water volumes of about half the size (43,000 vs. 83,000 m$^3$) develop green fringe areas over two orders of magnitude larger (100,000 vs. 900 m$^2$). This is probably because the water-bodies act as inert ponds, since the steeply sloping sides cannot supply the amount of water necessary for the progression of vegetation, quickly moving from a subaquatic state (permanent waterlogging) to a state of permanent dryness. On the Atlantic watershed, something similar can be observed, but although the water volume stored by overdeepening is greater than on the Mediterranean watershed (65,000 vs. 43,000 m$^3$), the extent of the associated green fringes decreases substantially (35,000 vs. 100,000 m$^3$).

Twelve water-bodies on this massif are associated with nine moraines—the Corral del Veleta moraine (mean altitude 3097 m) has four water-bodies, two of which (the highest and closest together) probably have a hydrogeological connection. However, we shall consider this group of water-bodies as a whole. The mean altitude of these nine moraines is 2979 m, distributed between 2743 m and 3097 m asl, five of them at over 3000 m. Six moraines have been counted on the Mediterranean watershed, with a mean altitude of 3001 m and an altitude range of 222 m. By contrast, the Atlantic watershed has three moraines at a mean altitude of 2906 m, with an altitude range of 354 m. Although there

are not enough moraines in the region for us to make definitive comparisons between the watersheds, we nonetheless observe that these altitude differences run parallel to the asymmetry of the slopes on each watershed (Figure 11)—the parts with less steep slopes (20.5% on the Mediterranean and 31.4% on the Atlantic watersheds) facilitate the installation of water-bodies, especially if they are linked to moraines. We therefore find this data set to be coherent with the hydrological behaviour of this massif but not with its xericity, and the answer to this paradox probably lies in the geological evolution of this environment in the short to medium term, in particular, with uneven uplift of individual blocks in the massif.

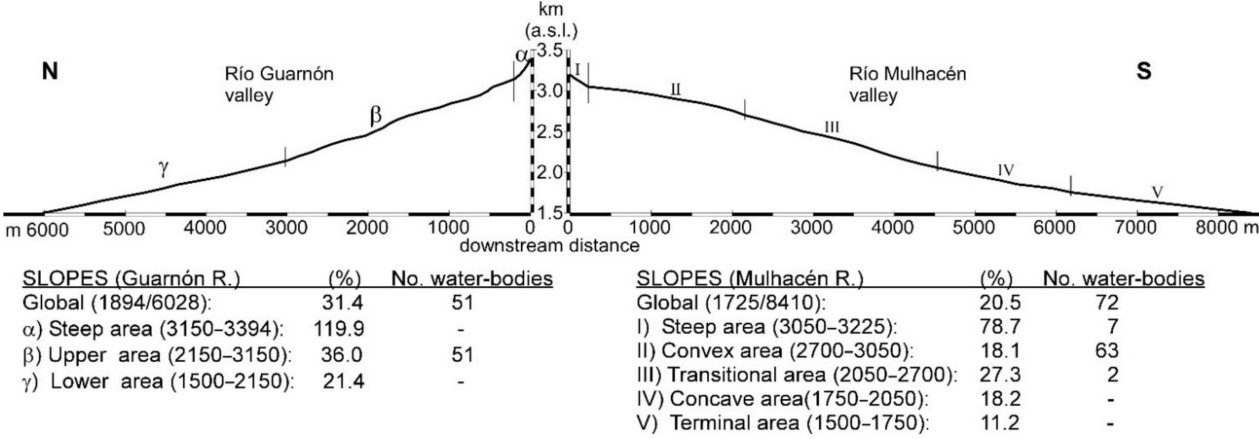

**Figure 11.** Geomorphological cross section along the central axes of two representative valleys: Guarnón River (Atlantic watershed) and Mulhacén River (Mediterranean watershed). The table at the bottom shows the general slope of each river and its different slope stretches (Greek and Roman symbols), and the number of water-bodies in each stretch. Transect has a general N–S direction. The numbers in brackets indicate, for the global slope, the quotient between altitude/length and, for other areas, the range of altitudes at which the slope is defined. Vertical scale = horizontal scale.

### 4.2. Water-Bodies Caused by Landslide

Table 1, part A shows that 25% of the water-bodies catalogued were caused by land-slides, and although they are not very numerous, they cannot be overlooked. The two watersheds present a similar number of water-bodies. They are located at a mean altitude of 2842 m, with a broad altitude range (3180—2498 m asl, Figures 8C,D and 9B), which is significantly different to the preceding category.

The Laguna de la Mula is a water-body with representative characteristics. It is located at the lowest altitude of all those in the inventory (2498 m asl), occupying the bottom of a small, slightly endorheic hollow (Figures 7 and 8C). The highest point of this hollow is 2756 m asl, and its distal extremes are 1.71 km apart on a north-facing slope. It lies, therefore, on the snowline of Sierra Nevada, where repeated freezing and thawing weathers the rocks throughout much of the year. This explains why the displaced material is loose, sandy and not cohesive, with no outcropping of bedrock, although the displaced mass contains occasional "floating rocks". An ancient residual erosive surface is found at the highest part, from which occurs towards the Dílar River flowing round the base. Landslide morphologies can be detected quite distinctly (Figure 12), for which recent activity is clearly seen in two lobes of frontal displacement to the north. The lower of these diverts the present course of the Dílar River, and the other overlaps on a small meadow at mid-slope (see the asterisks in Figures 8C and 12). Although the characteristics of the displaced material allow us to classify this as a debris flow, it could also be a rotational displacement where the presence of hypothetical breakage planes make it similar to the model in Figure 6B. The hollow containing this water-body is not therefore erosive sensu stricto, but it rests on an inclined plane (confined within this small hollow) and limited by shear planes (slip planes). For this reason, the water in this hollow is shallow (0.50 m). The surface area affected by

landslide is 186 ha, and the head scarp is over 10m in height. The La Atalaya and Peñón Negro water-bodies are other similar cases, although the first landslide is extensive, lying against the peak line (3000 m), and the second is smaller.

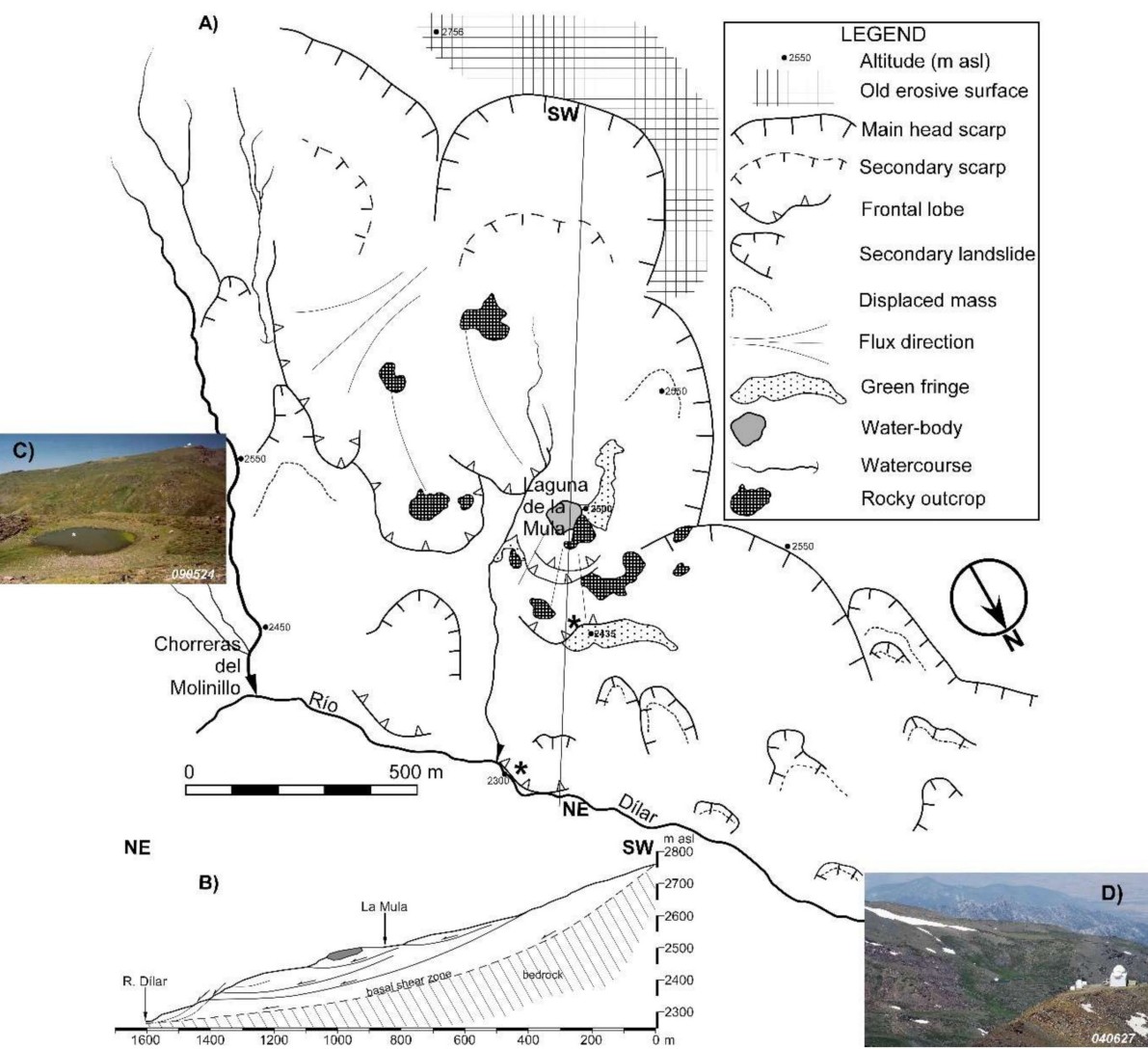

**Figure 12.** (**A**) Landslide mapping of Laguna de la Mula. Black asterisks indicate two fronts of this landslide, at two different levels: the higher overlaps a meadow, and the lower displaces the Dílar riverbed. See also Figure 6C. (**B**) Cross-section of a landslide. (**C,D**) Images obtained, respectively, in periods of drought and thaw. White asterisks indicate the position of the water body.

Landslides into valley bottoms can also hinder watercourses, thus damming the water to form small lakes. In these cases, the displaced mass usually consists of heterogeneous debris, and we have classified them as debris flows, although they can differ from the model schematised in Figure 6B. This is the case of the Río Lagunillos and Charcas de Peñón Negro water-bodies. Further observations could be made on the context in which these cases occur, but they are here omitted for the sake of brevity.

Finally, if landslides consist of masses of rocks, they can also create water-bodies if they come to rest on impermeable substrata (bedrock, in our case). The mean water depth in such cases is 1.1 m. Although there is abundant rocky debris in Sierra Nevada, this type of water-bodies is not common, because the debris is highly permeable and therefore unsuitable for water retention. Examples of this type of water-body are the Laguna de los Geólogos (Figure 8D) and Puesto del Cura.

These processes occur as a result of the de-structuring of the rocky massif by gelifraction and/or tectonic activity, which are responsible for erasing many features of glacial modelling.

*4.3. Water-Bodies and Tectonics*

Tectonic studies of the Sierra Nevada antiform mainly analyse the deformation in the massif from the viewpoint of orogenic phases, their intensity and evolution. They are, therefore, wide-range studies with a very different aim to that of the present study. However, at present, there are few studies on the neo-tectonics of the massif. More specifically, we need to have a better understanding of its evolution throughout the late Quaternary-Holocene—the periods of glacial modelling we can best analyse. The inherent difficulties in the undertaking of such studies are the significant lithological monotony of the schists [45], their ease of weathering [68] and the coverings caused by the latter, questions that need better understanding in the specific context of Sierra Nevada [69,70]. However, at present, we can find signs of neo-tectonic activity such as areas of intense fracturing, fractured and displaced sheepback rocks, large escarpments and hanging glacial valleys. In fact, medium to large landslides lie on the blurry boundary with tectonic processes.

It could be considered that the tectonic elevation rate also affects the geomorphological modelling of Sierra Nevada [43]. If this rate is higher than that of denudation by erosion, a positive relief is formed, characterized by steep inclines and deep gullies such as those found in this massif. This causes accommodation by both normal faults and by the slope instability phenomena. Some authors [31] considered that landslides are the main denudation modes in tectonically active mountain ranges. Sierra Nevada is a young massif with high elevation rates [32,33] and should therefore offer examples of landslides. This issue requires further attention. The water-bodies studied here act as detectors of this activity on the 2480–3200 m strip along which they are concentrated and can be clearly observed along the intermediate strip (mean altitude 2842 m asl). It should be noted that it is a matter of debate whether the tectonics play an active or passive role in controlling the triggering and progress of landslides [71–73]. Nonetheless, the type II group of water-bodies is by definition linked to both internal (tectonics) and external (associated instability) geodynamic processes. This suggests that these water-bodies are of unstable nature, with a potentially shorter life than those of type I.

**5. Conclusions**

The 123 water-bodies in Sierra Nevada belong to three categories: (I) those originating in glacial processes (overdeepening and moraine dams) represent 72% of the total, are located at a mean altitude of 2918 m asl, and have a mean water depth of 1.20 m; (II) those related to slope instability (debris-flow and rock fall) make up 25% of the total, located at a mean altitude of 2842 m asl, with a mean water depth of 0.84 m; and (III) those making up 3% of the bodies are attributed to anthropic and unknown causes. These water-bodies are mostly linked to watercourses.

The volume of water stored in the Mediterranean high-mountain water-bodies of Sierra Nevada is approximately 215,000 m$^3$, of which 90% (194,000 m$^3$) is contained in water-bodies of glacial origin. The type II water-bodies contain 20,000 m$^3$ (9%) of the total water, and the type III bodies are irrelevant (<1%). The depths of the water-bodies are significantly different in each case.

The green fringes surrounding the studied water-bodies have a total surface area of approximately 186,000 m$^2$, of which 136,000 m$^2$ (73%) are located around type I water-bodies and the remaining 49,000 m$^2$ (26%) are around those of type II. The third category is of no interest in this regard.

These figures, referring to the natural environment of Sierra Nevada at altitudes >2500 m asl, also reflect the greater hydrological importance of the water-bodies on the Mediterranean watershed over those on the Atlantic watershed, which is surprising because

of the higher xericity of the former. The cause of this hydrological imbalance between watersheds seems to be unrelated to the formation processes of the existing water-bodies and could be related to the different inclinations of the two slopes, which is controlled by tectonic activity. Consequently, the water-bodies in areas near the peak line (mean altitude 2918 m asl) are highly dependent on the glacial processes causing the hollows where they lie. In addition, the slope instability processes caused water-bodies located mainly at intermediate altitude (mean 2842 m asl) and represent areas of tectonic weakness or debris accumulation. These water-bodies would be unstable in nature, probably with a shorter life span than those caused by glaciers. However, the importance of the instability phenomena at lower altitudes lies in the significant role they must have played in the emission of volumes of eroded material. This flow of material would probably have taken place via the present drainage network. It would also be of interest to specify the variations in water quality in the streams linked to the water bodies studied, as they could have disparate behaviour in the June to November period, resulting from processes of thaw and drought.

In short, the present stage of evolution of this landscape determines that the most important water-bodies are those of glacial origin both in number and by water volume and stability.

**Author Contributions:** Both authors have contributed to the conceptualization, methodology, formal analysis, investigation, resources, writing—original draft preparation and writing—review and editing. Both authors have read and agreed to the published version of the manuscript.

**Funding:** This research received no external funding.

**Institutional Review Board Statement:** Not applicable.

**Informed Consent Statement:** Not applicable.

**Data Availability Statement:** The data presented in this study are original and available on request from the corresponding author. Climatic data were obtained from the web, as indicated in the text.

**Acknowledgments:** Field work was carried out from 2000 to 2018 in the setting provided by the activities of research group RNM-208 (Junta de Andalucía). All images shown were obtained by the authors except those included in Figure 6, which correspond to Google Earth Digital Globe imagery to which we are grateful. Similarly, we express our gratitude to Google Maps and Iberpix, which made it easier to find the exact location of all points studied and facilitated the cartography. We also wish to thank the anonymous reviewers and assistant editor Teodora Rusu for their very thorough reviews and helpful comments on the manuscript.

**Conflicts of Interest:** The authors declare no conflict of interest.

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
