# Peer review of "Formation Patterns of Mediterranean High-Mountain Water-Bodies in Sierra-Nevada, SE Spain"

_water, doi:10.3390/w13040438_

Round 1

Reviewer 1 Report

The manuscript describes the mapping and cataloguing of water bodies in Sierra Nevada Massif. the proposed topic is of high interest in the current climate change scenario.

The paper organization should be improved:

The introduction section is too generic and, for example, lacks in-depth analysis concerning the techniques used for water bodies mapping. E.g.:

  • Soti, V., Tran, A., Bailly, J.S., Puech, C., Seen, D.L. and Bégué, A., 2009. Assessing optical earth observation systems for mapping and monitoring temporary ponds in arid areas. International Journal of Applied Earth Observation and Geoinformation11(5), pp.344-351.

Moreover, an assessment of the water characteristics should be added as its chemistry could be variable and therefore the quality and, potential, usability:

  • Colombo, N., Gruber, S., Martin, M., Malandrino, M., Magnani, A., Godone, D., Freppaz, M., Fratianni, S. and Salerno, F., 2018. Rainfall as primary driver of discharge and solute export from rock glaciers: The Col d'Olen Rock Glacier in the NW Italian Alps. Science of the Total Environment639, pp.316-330.
  • Thies, H., Nickus, U., Tolotti, M., Tessadri, R. and Krainer, K., 2013. Evidence of rock glacier melt impacts on water chemistry and diatoms in high mountain streams. Cold Regions Science and Technology96, pp.77-85.

Methods should be improved by adding an accuracy assessment. Authors describe mapping procedures without specifying their reliability and consequently, the area and volume estimates could be affected by several types of errors.

Results are well organized. I would only add a time series depicting the water body evolution during the investigation period. It would be interesting to compare it with meteorological variables. It could be an alternative point of view to analyze water body behaviour in the different sectors of the study area.

Discussion is well organized concerning the subsection related to glaciers. On the other hand, the other ones are too generic.

I also added some minor comments in the attached PDF.

Author Response

Reviewer #1:

First of all we should express our gratitude to the Reviewer by your work and rigorous comments which improve this research and reaffirm us in the performed work.

The manuscript describes the mapping and cataloguing of water bodies in Sierra Nevada Massif. the proposed topic is of interest in the current climate change scenario.

The paper organization should be improved:.

Point 1. The introduction section is too generic and, for example, lacks in-depth analysis concerning the techniques used for water bodies mapping. E.g.:

Soti, V., Tran, A., Bailly, J.S., Puech, C., Seen, D.L. and Bégué, A., 2009. Assessing optical earth observation systems for mapping and monitoring temporary ponds in arid areas. International Journal of Applied Earth Observation and Geoinformation, 11(5), pp.344-351. https://doi.org/10.1016/j.jag.2009.05.005

Moreover, an assessment of the water characteristics should be added as its chemistry could be variable and therefore the quality and, potential, usability:

Colombo, N., Gruber, S., Martin, M., Malandrino, M., Magnani, A., Godone, D., Freppaz, M., Fratianni, S. and Salerno, F., 2018. Rainfall as primary driver of discharge and solute export from rock glaciers: The Col d'Olen Rock Glacier in the NW Italian Alps. Science of the Total Environment, 639, pp.316-330. https://doi.org/10.1016/j.scitotenv.2018.05.098

Thies, H., Nickus, U., Tolotti, M., Tessadri, R. and Krainer, K., 2013. Evidence of rock glacier melt impacts on water chemistry and diatoms in high mountain streams. Cold Regions Science and Technology, 96, pp.77-85.”

We agree to a large extent with the opinion of the Reviewer. But it must be borne in mind that the subject studied has many facets and, consequently, many approaches converge in it, so that not all can be addressed. So, we give in this section a general view of this topic and originally we selected the closest topics to the central idea (patterns in formation), while others were forsaken not only for space reasons. However, it seems appropriate to complete this Introduction with some of the ideas and References indicated by the Reviewer (see Ls94-97,99-101,573-579). Even so, we consider that the length of the paper will have reasonable limits.

Point 2. “Methods should be improved by adding an accuracy assessment. Authors describe mapping procedures without specifying their reliability and consequently, the area and volume estimates could be affected by several types of errors.”

Ok. We have considered three points in this regard. Firstly, the calculations of the volumes are always approximate: this is why it makes no sense to give excessively precise values; in fact unnecessary precision is indicative of unreliable data. Therefore, our calculations state maximum volumes, as is expressed in Ls182,187-188,197-198 because we have used indicators of maximum levels, which are reasonable for the objectives proposed. Secondly, these volumes were compared with internal reports (measurements of 20 water-bodies, 2009), obtained by bathymetry based on GPS stations: these methods cannot be systematically applied to all inventoried water-bodies, at least at present, for logistical reasons. The comparison of our data with these previous studies allowed us to ensure the preceding point (R2=0.95). And thirdly, we have used several formulas to calculate these volumes (Ls189-191), depending on the different morphology of the bottoms of the water-bodies. Therefore we have added a concise commentary about the second point (Ls191-197).

In addition to the previous point, in our field visits we have enclosed each water-body in a rectangle whose two dimensions were compared with those obtained by aerial photography, obtaining a very good correspondence. In these cases the plane of the water body is practically horizontal, and the distortion was minimal.

Point 3. “Results are well organized. I would only add a time series depicting the water body evolution during the investigation period. It would be interesting to compare it with meteorological variables. It could be an alternative point of view to analyze water body behaviour in the different sectors of the study area.”

In principle the variety of types of these water-bodies is very large, and a specific monitoring of each water-body is impossible in our case. For this purpose, human and material resources would be required that are beyond our means. This point could be the subject of future planning for water research in this area. But at present, the only time series is that of a reservoir on the Genil River, at low altitude, and therefore is not useful for this purpose.

Point 4. “Discussion is well organized concerning the subsection related to glaciers. On the other hand, the other ones are too generic.”

Yes, this is true. This is so because the primary objective was to complement and highlight the aspects that converge in the formation of these water-bodies. At present this approach in this area is poorly known, in spite of the geomorphological studies performed. Regarding the influence of landsliding in the formation of these water-bodies, this is the first time that this presence has been recognized at this site, precisely because they are in remote places. But this topic requires further research, which is far from the approach of this paper as we have indicated. For this reason, we are working at the present on an exclusive investigation on landslides in this area, and therefore here we have only detailed some paradigmatic cases.

I also added some minor comments in the attached PDF:

Point 5. Figure 1 “(shrink the Mulhacén Peak label):”

Ok, we have improved this Figure 1 (new Figure).

Point 6. Line 51 "(relationships of this sentence with the manuscript topic):”

The objective of this sentence is to reveal the action of instability phenomena in this area, with easily observable examples. We have added a connector to improve this apparent conceptual disconnection (L90).

Point 7. Line 62 “(shallow phenomena? In this context?):”

Ok, we have deleted “shallow” although this is a relative term. We will treat this topic in a posterior article, specifically concerning landslides (L64).

Point 8. Line 69 “(dam effect):”

Ok, this has been clarified (L71).

Point 9. Line 80 “(Mountains Agenda and SDGs):”

Ok, this has been expanded a bit (L99-101). Thank you for your collaboration.

Point 10. Line 130 (Delete authors):”

Ok, this was a lapse (L151).

Point 11. Line 160 “(Every year?):”

Yes, but the water-bodies are covered in snow for a large part of the year. We have added a short clarification (L177).

Point 12. Line 170 “(raw or orthorectified?):”

We have used raw images, but these images come from Google Earth, magnified to the maximum size allowed by the computer screen and taking care that the plane of the image is an orthogonal projection. As commented before the details obtained by this method are very good, as well as their precision. This has been specified in the text (L187).

Point 13. Line 180 “(handheld …?):”

Ok, we have specified this point (L204). We created a database with these data, which will probably be published in the future.

Point 14. Line 190 “( geomorphological map and landslides):”

No, the published Sierra Nevada Geomorphological Map does not include landslides, and many water-bodies are missing in the present published bibliography.

Point 15. Line 199 “(remove reference):”

Ok (L225).

Point 16. Line 232 “(name):”

Yes, this water-body is new: until now it has not appeared on any map. Access to it is difficult.

Point 17. (Table 1: word Glaciar):“

Ok, lapse improved.

Point 18. L297 “(anthropic modifications):“

Ok. We have detailed some of them, although at present the conservation is quite good (Ls337-338).

Point 19. Ls317-318 (heading Figure 7):”

Ok, it was a superfluous clarification (L358). Now is Figure 8.

Point 20. L339 “(I suppose a writing mistake):”

Both cases have the same “±” sign. This was a change caused by the system (L372).

Point 21. Ls387-394 (transfer paragraph to methods):”

Ok (Ls231-238).

Point 22. L419 (picture Figure 10):”

Ok, picture added (now Figure 11).

Point 23. Ls433-439 (add References):”

Ok, we have added four References (Ls466-468).

Reviewer 2 Report

see attached file

Author Response

“Reviewer #2:

First of all we should express our gratitude to the Reviewers by your work and rigorous comments which improve this research and reaffirm us in the performed work.

Text: general considerations

 Point 1. “In this work, the authors have identified a number of unevenly distributed natural water-bodies of various sizes occupying the study area. Given that the special issue discusses the specific theme of hydrology, geomorphology, and ecology of intermittent rivers and streams, it would be necessary if the authors would illustrate the possible links with stream that may originate from these high-altitude water bodies, as seen for example in figure 6a and 6b. This would be useful to highlight importance of these streams for aquatic habitats, as requested by the special issue.

I believe this link is important in order to highlight the strategic role these water bodies can play in controlling the aquatic environments related to them and thus meet the objective of the special issue.”

Ok. We have added a paragraph in the Introduction and several comments distributed throughout the text (L35,77-90,102-103,249-252,256,315-317,511-518). We should note that Figure 5 contains all the high-mountain streams linked to water-bodies, and we believe that it constitutes a good link in the sense indicated by the Reviewer.

Some additional illustrative images (Figure 2), especially related with green-fringes, have been added.

Point 2. “I think it would be useful to include a geological-geomorphological map of the study area.”

There is a Geomorphological Map (2002), published by the Andalusian Government (see reference 10). Therefore we consider including such Map could be plagiarism. On the other hand, this map has a great format, and it is very detailed, although landslides are omitted. A reduction of this Map would entail loss of much information, as for instance happens with Figure 6, which is only a sketch.

Point 3. “This work has highlighted the strategic role of these areas as important water reservoirs in a typical high mountain environment. The authors say that there are few weather stations in the study area. Has this impeded the ability to do a more detailed climatic reconstruction?”

Unfortunately this is the case. Sometimes new data appear corresponding to new stations, but there are very few of these weather stations, with a very specific purpose. At present, new research is being undertaken in this sense, but their data are not so detailed that those of weather stations: we refer to palinological research, studying the buried sequences linked to evolution of the vegetation of this area, but they also record relatively short periods of time, no more than the beginning of the Holocene, in the best of cases.

Point 4. “The few weather stations present, allow to make a more detailed climatic framework of the study area? Are there historical series of precipitation (rain and snow) and temperature to evaluate the natural recharge in the study area and therefore the possible impact of climate change on these high-altitude environments?”

We consider that, at present, this is not possible. This is the problem of the mountain areas, or at least in this mountain area. For this reason botanists have created a bioclimatic classification system, based on the distribution of the living species in particular climatic conditions. Some monitoring of runoff stations (linked to water reservoirs) have some data, but all they are at low altitude. The original interest of these stations is the water management (agriculture and potable water supply). Many times these data sequences have deficiencies. In addition, and specifically in the surroundings of Sierra Nevada, there are recent reservoirs whose climatic and runoff data are incipient, if any, and always they are at low altitude.

For these reasons, I suggest to reconsider the article after major revision.

Text: detailed considerations

Point 5. Line 93: “There are few weather stations, but the mean precipitation recorded varied from 710 mm at the….. Question: annual mean precipitation?”

Yes, these values are mean annual precipitation. We have improved this term (L113).

Point 6. Lines 181-182: “The data base is also available on software GIS type?”

We have only organized this database in a worksheet, which includes numerical and alfanumerical data of the water-bodies.

Point 7. Line 211: “123 water-bodies have been catalogued in the study area, how many of these water-body are fall within the National Park?”

They all fall within the National Park. This protective figure is the most restrictive because this area is mainly rich in endemisms.

Point 8. Line 304: “In the study area have been mapped habitats according to Natura 2000 standards or similar? If so, it would be interesting to understand the link between aquatic habitats and green fringes surrounding the studied water-bodies and streams and therefore of their ecological and ecosystem value.”

We have knowledge that there are studies according to Natura 2000 standards. However, the specific study of green fringes is outside of our objectives because it is not our specialty: this topic is more related to ecology or eco-physiology.

Point 9. Line 409: “Do the surfaces that control sliding have any relationship with the main tectonic lineaments present in the study area?”

We do not know exactly because frequently there are convergences between sliding surfaces and tectonic alignments and the problem is deciding what is the cause and what is the effect. The better solution is to find a set of features that support one or the other, but this is made difficult by the glacial action which blurs the original structures, especially in the old landslides.

Figures and tables

Point 10. Figure 1: “Is it possible to insert altitude points and/or contour lines?”

We prefer not to touch the hillshade area because we can lose the sense of volume. In fact another Reviewer has advised deleting any label within this area to preserve this effect. However, we present now a new Figure 1.

Points 11, 12 and 13. “Figure 2: Is it possible to report a scale in the different images (a, b, c and d)?

Figure 3: Is it possible to report a scale in the different images?

Figure 6: Is it possible to report a scale in the different images?”

These images are landscapes and therefore these perspectives are in conical projection, and consequently a scale (as a rule or an object superimposed) it would be misleading. We understand your proposition, but for the sake of rigor we prefer not to touch these images.

Point 14. “Figure 10: Is it possible to insert altitude points and/or contour lines?”

Yes, we have inserted some altitude points (see the new Figure 11). But we prefer not to insert contour lines because the figure would be confusing.

Point 15. “Table I: Sixth Column from the left, the sum is 215063 and not 215062. This happens for other sums, why?”

This is a question of decimals and it doesn’t matter at the scale of this study. This table comes from a worksheet and automatically rounds up the numbers. It has been improved.

References

Adjustment to the journal standards is needed. See Reference List and Citations Style Guide for MDPI Journals.

For example:

Point 16. “a) Insert a comma after names and replace page separator (-) with –

Ok, we studied the Reference List and Citations Style Guide for MDPI Journals and we have carried out the changes proposed.

Point 17. “b) Change the format of citations (5) (6) as follows:

Author 1; Author 2; Author 3; etc. Title of the contribution. In Title of the Book, Edition (if available); Editor1, Editor 2, Eds.; Publisher: City, Country, Year; Volume (optional), pp. Pages (optional), DOI or ISBN XXXXXX-XXX-XXX-X (if available).”

Ok, we have performed the indicated changes.

Point 18. “c) In reference (3) replace Geomorphology with Geomorphology and Natural Hazards.”

Ok, change made (L532).

Point 19. “d) Add DOI in the following references:

(3) https://doi.org/10.1016/B978-0-444-82012-9.50012-8”

Ok: thank you very much for your contribution. Addition completed (L533).

Point 20. “e) Check authors sequence (reference 11). I have found: Deglaciación reciente de Sierra Nevada Repercusiones morfogénicas, nuevos datos y perspectivas de estudio futuro. David Palacios Estremera ; Lothar Schulte ; Ferran Salvador Franch; José Juan de Sanjosé Blasco ; Alan Davis JamesAtkinson ; Antonio Gómez Ortiz Is the same?”

These articles are different. We have added the DOI, and the title now contains its original English title (Ls551-552).

Point 21. “f) check reference citation (13)”

Ok: this reference has been checked and is correct, and has not DOI.

Point 22. “g) (14) standardize authors sequence format, Surname, Name.;”

Ok: It has been checked and in addition we have changed the data of volume and pages (L558-559).

Point 23. “h) Only the following DOI have reported a problem: Reference (23), see screen shot below (31-12-2020). SCREEN SHOT”

Ok: the Institutional web pages are very unstable for many reasons. This web is now different because political changes have caused changes in the Name of the Institution: formerly it was “Ministry of Agriculture and Fisheries, Food and Environment (Ministerio de Agricultura Pesca y Alimentación, MAPAMA in Spanish) and now it is the more complex “Ministry for the Ecological Transition and Demographic Challenge” (Ministerio para la Transición Ecológica y el Reto Demográfico, in Spanish). We do not know how to solve this question. This is why our opinion would be to do without them: We have no words to express our anger. Please, help us to correct this deficiency.

Point 24. “i) References (39): These DOI (doi.org/10.1016/j.quascirev.2011.03.005, 2011) cannot be found in the DOI System. All the others DOI are ok.”

Ok. Mistake corrected (the last year should be removed). This reference has now the number (42), see L625.

Point 25. “j) couldn't find article (22)”

This reference really is “Rev. Cuater. Geomor” (also include the first word). We have corrected this mistake. Does not have DOI. This reference is now (25), see Ls584-585).

Point 26. “k) for the references (24, 25, 26 and 41), is it possible to indicate a portal where to download the maps?

Is this correct?: https://info.igme.es/cartografiadigital/geologica/Magna50.aspx”

As we have seen above, the Institutional webs have serious risks of change. However, the indicated portal works well and it is easy find the maps of the specified references.

Point 27. “l) Add DOI at reference (36): https://doi.org/10.21138/bage.1539”

Ok, thank you very much for your contribution. Change made (L616).

Point 28. “m) Add DOI at reference (38): https://doi.org/10.1016/S1040-6182(02)00048-4”

Ok, we are very grateful for this support. Change made (L622).

Point 29. “n) Rewrite the bibliographic citation (53) as follows:

Darrel A. Swift, Simon Cook, Tobias Heckmann, Jeffrey Moore, Isabelle Gärtner-Roer, Oliver Korup, Chapter 6 - Ice and Snow as Land-Forming Agents, Editor(s): John F. Shroder, Wilfried Haeberli, Colin Whiteman, Snow and Ice-Related Hazards, Risks and Disasters, Academic Press, 2015, Pages 167-199, ISBN 9780123948496,

Ok: thank you very much for your contribution. We have unified style with that of the Reference 6 (Chapter1) and we have add the equivalent DOI (Ls631-634).

Round 2

Reviewer 1 Report

The authors improved substantially the manuscript. There is only one weak point in the methodology as the lake mapping is affected by high uncertainties. The authors state that their data are in good agreement with previous ones. I recommend adding an XY plot showing this agreement. Additionally, I'm not sure that R² is the best way to assess the agreement; a correlation coefficient would be more appropriate. In any case, a statistical significance value should be added.

Concerning field measurement, with the information provided, I can assume that accuracy is low or very low as the phase is carried out with manual means and/or raw GPS data. Authors, in fact, states that all the results are estimates of volume. I suggest further stress on this point in order to avoid misleading the reader.

Author Response

Reviewer 1:

The authors improved substantially the manuscript. There is only one weak point in the methodology as the lake mapping is affected by high uncertainties. The authors state that their data are in good agreement with previous ones. I recommend adding an XY plot showing this agreement. Additionally, I'm not sure that R² is the best way to assess the agreement; a correlation coefficient would be more appropriate. In any case, a statistical significance value should be added.

Ok. All this has been added: certainly this plot is the best manner to show these relationships. Furthermore, we agree with you that it is better to use Pearson’s coefficient (r), as well as add the p-value. (See Figure 5).

Concerning field measurement, with the information provided, I can assume that accuracy is low or very low as the phase is carried out with manual means and/or raw GPS data. Authors, in fact, states that all the results are estimates of volume. I suggest further stress on this point in order to avoid misleading the reader.

We have stated in the previous reply that any measure carried out in nature is an estimation. However, probably the estimation of the water volumes “closest to the truth” is that obtained with GPS, although this is really hard to put into practice: therefore, we have used another method which has been contrasted with this “truth”. Consequently, once favorably contrasted, we have applied a feasible method to the universe of water-bodies of this place. Plots now included in the new Figure 5 were performed a long time ago, as a result of previous considerations in the sense indicated by the Reviewer, but were not included in the first version because we thought that they are routine, although, certainly, give security about what has been done. Your authoritative opinion prompts us to include your suggestion. Thanks for your advice. (See Ls188-191,194,197-199).

All new changes have been written in blue in the Manuscript

Reviewer 2 Report

I thank the authors for their answers and for the work done to improve the text.

Author Response

Reviewer 2:

I thank the authors for their answers and for the work done to improve the text.

Ok. Thank you very much for your support.

Round 3

Reviewer 1 Report

The manuscript is suitable for publication in its current state

Academic Editor Notes

The authors have done a good job of addressing reviewer comments in the current revision. However, one lingering issue remains. As a follow up to Reviewer 2’s comment below, I think a little more work can be done to better link this study to the objective of the Special Issue, which I think can be done with a few more lines of text in the abstract, introduction and discussion/conclusion. This includes providing a sentence/statement in the abstract that provides justification for the why this study is a reasonable fit for the issue. Specifically, this sentence could highlight the impermanence, fragility, and relatively unknown understanding of these features as well as some phrase to justify their importance, as part of the growing body of research that document intermittent freshwater features.

Ok. We have modified and expanded the first sentence of the Abstract. We believe the changes are well placed here in response to your suggestions (Ls11-17). We have included new sentences in Conclusions (Ls506,524-526).

In addition, justification in the main body of the text in the Introduction is very brief and limited to a single reference (Colombo et al., 2018) that documents rainfall as a driver of export from rock glaciers, but no mention of literature in streamflow permanence/intermittency. Please add to this paragraph language that more strongly links this study to the special issue objectives.

Ok. We have also extended the Introduction a little (Ls43-48) and added ten References (Ls542-566).

Other modifications of interest:

We have updated Reference 40, although this type of “official” reference will always be unstable (Ls638-639).

We have also observed that “Author Contributions” on the web do not coincide with the content of this point in the text (Ls532-533). We do not know what the effects of this difference might be.

Finally, consequently with these changes all the References of the manuscript have been reordered.
